# KCC2 overexpression prevents the paradoxical seizure-promoting action of somatic inhibition

Vincent Magloire [1], Jonathan Cornford[1], Andreas Lieb [1], Dimitri M. Kullmann [1] & Ivan Pavlov[1]

Although cortical interneurons are apparently well-placed to suppress seizures, several recent reports have highlighted a paradoxical role of perisomatic-targeting parvalbumin-positive (PV+) interneurons in ictogenesis. Here, we use an acute in vivo model of focal cortical seizures in awake behaving mice, together with closed-loop optogenetic manipulation of PV+ interneurons, to investigate their function during seizures. We show that photo-depolarization of PV+ interneurons rapidly switches from an anti-ictal to a pro-ictal effect within a few seconds of seizure initiation. The pro-ictal effect of delayed photostimulation of PV+ interneurons was not shared with dendrite-targeting somatostatin-positive (SOM+) interneurons. We also show that this switch can be prevented by overexpression of the neuronal potassium-chloride co-transporter KCC2 in principal cortical neurons. These results suggest that strategies aimed at improving the ability of principal neurons to maintain a trans-membrane chloride gradient in the face of excessive network activity can prevent interneurons from contributing to seizure perpetuation.

---

[1] Department of Clinical and Experimental Epilepsy, UCL Institute of Neurology, University College London, London WC1N 3BG, UK. Correspondence and requests for materials should be addressed to V.M. (email: v.magloire@ucl.ac.uk) or to I.P. (email: i.pavlov@ucl.ac.uk)

Poor understanding of the cellular and circuit mechanisms underlying the generation and maintenance of seizures continues to limit our ability to treat epilepsy successfully[1]. Seizures are readily precipitated by experimental disinhibition, and a loss of interneurons has been documented in chronic epilepsies, providing support to the general principle that impaired interneuron function is a key ictogenic factor. Indeed, failure of an inhibitory restraint has been suggested to underlie seizure initiation and propagation in both animal models and human studies[2–4]. This has justified the enhancement of GABAergic neurotransmission as an anti-epileptic strategy. However, it has recently become apparent that the role of GABAergic interneurons in ictogenesis is more complex. Several studies have shown that GABAergic signalling can sometimes paradoxically contribute to epileptiform discharges[5–9], and some have even suggested that activation of the interneuron network alone can elicit epileptiform activity[10].

The controversy regarding the role of GABA-mediated neurotransmission in the generation of epileptiform activity[11,12] has recently received further impetus from several optogenetic studies, in which photostimulation of interneurons paradoxically promoted ictal discharges[13–16]. The translational relevance of these observations, made in acute brain slices, is yet to be established, calling for conclusive in vivo evidence to clarify the roles of specific types of interneurons in ictogenesis. Surprisingly, both optogenetic activation and inhibition of perisomatic-targeting parvalbumin-positive (PV+) cells, the population of interneurons that provide the main source of somatic inhibition, have been reported to reduce seizures in vivo[17,18]. In both studies, however, the location of photostimulation with respect to the seizure initiation site was undefined, making interpretation of these results difficult in light of potentially different roles of (peri-) focal and distant interneurons[15].

Several mechanisms could potentially account for pro-seizure effects of interneuron activation. Among these, a switch of GABAergic signalling in principal cells from inhibitory to excitatory is a plausible candidate[19,20]. Such a switch in the action of GABA following intense network activity leads to the prediction that the role of interneurons in ictogenesis should depend critically on the timing of their recruitment during a seizure. Promoting interneuron firing early in a seizure should therefore suppress it, but the same manipulation later in a seizure when postsynaptic chloride accumulation has occurred should facilitate ictal activity. To test this prediction, we used automated seizure onset detection in combination with closed-loop optogenetic stimulation in a focal model of cortical seizures in awake behaving mice. We show that photo-depolarization of local PV+ interneurons switches from an anti-ictal to a pro-ictal effect over the first few seconds of seizure initiation. We further show that the pro-ictal action is not shared with dendrite-targeting somatostatin-positive (SOM+) cells, and can be abolished by over-expression of the potassium-chloride co-transporter KCC2 in principal cortical neurons. The results provide compelling evidence for a breakdown of chloride homeostasis as central to the perpetuation of seizures.

## Results

**Focal neocortical seizure model**. To induce focal cortical seizures in awake mice, we performed acute injections of pilocarpine (3.5 M, 0.2–0.4 µl) into the deep layers of the primary visual cortex, V1, through an implanted cannula (Fig. 1a). The same cannula was then used to position a laser-coupled optic fibre for photostimulation. Electrographic activity was monitored using an electrocorticogram (ECoG) electrode located above the injection site and connected to a subcutaneous wireless transmitter (see

Methods). The onset of epileptiform activity was usually characterized by the appearance of brief repetitive negative spikes, which gradually evolved into more complex seizure-like events. These ictal ECoG discharges lasted several seconds and had a distinct high-frequency component. Combined ECoG recording and video monitoring showed that episodes of electrographic ictal activity were often associated with clear behavioural manifestations (see Supplementary Movie 1). Towards the end of an experiment, recurrent seizures either gradually disappeared or, in some cases, developed into continuous epileptiform activity. Overall, as is typical for other epilepsy models, pilocarpine produced a spectrum of pathological epileptiform discharges of diverse lengths, frequencies and morphologies that varied over the course of an experiment and across different animals. Therefore, in order to automate the analysis and remove any potential bias as might arise from manual data selection, we trained a random-forest classifier[21] to restrict analysis to recordings that displayed discrete electrographic seizures. Briefly, features such as band-power and standard deviation (see Methods) were extracted from consecutive 10-s ECoG segments to distinguish between the various network states, namely: interictal bursting (brief 100–150 ms-long discharges; State 1), electrographic seizures (State 2), and continuous epileptiform activity (State 3) (Fig. 1b). The classifier was initially trained using a dataset of 740 ECoG epochs ($n = 7$ mice) assigned to States 1, 2 and 3, and then validated on 204 unseen traces from three different animals (f1-score: 0.93, Fig. 1c, d). The trained classifier was then used to categorize the 10 s of recording immediately preceding each photostimulation, and if it was not in State 2 (seizures), the trial was rejected. Furthermore, in order to exclude periods of ambiguous or mixed activity, only periods for which classifier confidence for State 2 was more than twice that of the second most likely state were included in the analysis.

**Continuous photostimulation of PV+ interneurons**. We first calibrated the optogenetic stimulation protocol using cell-attached recordings in acute cortical slices from mice conditionally expressing ChR2 in PV+ neurons (PV::Cre × Ai32). Photostimulation by 470 nm light evoked continuous spiking of ChR2-expressing cells (Fig. 2a). The 10 s-long light pulses ensured that interneuron depolarization exceeded the typical duration of seizures. Although some interneurons exhibited a decrease in action potential current amplitude towards the end of the light pulse suggestive of depolarization block (Fig. 2b, cell 3), we found that with the highest light intensities (12 mW at the objective; similar to those used in subsequent in vivo experiments at the tip of the optic fibre) at least 60% of PV+ interneurons reliably fired action potentials for the entire duration of the light exposure ($n = 10$). Voltage-clamp recordings from layer 5 pyramidal neurons in the visual cortex further confirmed that optogenetic stimulation elicited pure GABAergic currents (Supplementary Fig. 1a). Charge transfer through GABA$_A$ receptors in principal neurons evoked by continuous photostimulation was similar to that produced by intermittent light stimulation of matching duration, as per protocols used in previous studies[15,17] (Supplementary Fig. 1b).

We then asked if photoactivation of PV+ interneurons in vivo through an optic fibre positioned near to the site of pilocarpine injection affects the duration of seizures in awake behaving mice (Figs. 2c, 3a). We developed an automated system, in which light pulses were triggered by the onset of ictal discharges, and were separated by non-stimulation intervals of at least 20 s to allow collection of 10 s-long pre-stimulation and post-stimulation ECoG epochs (Supplementary Fig. 2a). Since all electrographic seizures in our model had a characteristic large negative sentinel

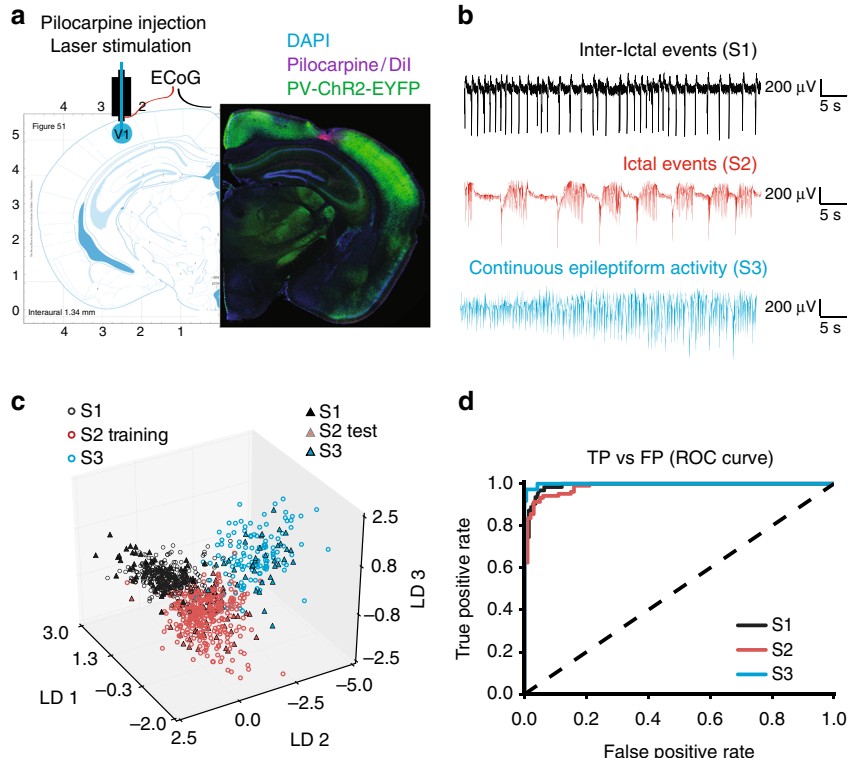

**Fig. 1** Acute epileptiform activity induced by pilocarpine injections in the mouse visual cortex. **a** Left: diagram showing the cannula guide location over the site of pilocarpine injection, ElectroCorticoGram (ECoG) recording and photostimulation in the visual cortex V1 area. Right: immunofluorescence of PV+ interneurons expressing ChR2-EYFP (green), cell nuclei staining DAPI (blue) and DiI co-injected with pilocarpine (magenta). Restricted DiI spread suggests that pilocarpine has a local action. **b** ECoG traces showing three different types of epileptiform activity induced by pilocarpine (States 1, 2, 3). **c** 3D plot of the linear discriminant analysis demonstrates network state separation on the training (open circles) and test (filled triangles) ECoG samples. Each linear discriminant (LD1, LD2, LD3) corresponds to a particular combination of weights of the 20 features extracted from ECoG traces. **d** Performance of the random forest network state classifier expressed as a Receiver Operating Characteristic (ROC) curve. TP true positives, FP false positives. The dashed line shows random allocation of events. Source data are provided as a Source Data file

spike at the beginning of a discharge, we used voltage threshold-crossing for on-line seizure detection. Photostimulation of PV+ interneurons profoundly reduced the average duration of seizures from 2.84 ± 0.5 s to 1.78 ± 0.41 s (50 trials in seven mice, one-way repeated measures ANOVA was used to compare seizures before, during and after photostimulation, $F$ (2, 12) = 11.6, post hoc Bonferroni correction, $p$ = 0.021). This suppression of pathological electrographic activity was only evident during photostimulation, as the average length of discharges returned to the pre-stimulation level immediately after the termination of the light pulse (3.33 ± 0.64 s, 50 trials in seven mice, one-way repeated measures ANOVA, $F$ (2, 12) = 11.6, post hoc Bonferroni correction, $p$ = 0.012; Fig. 2d, e). PV+ photostimulation did not affect the frequency of ictal discharges (pre-photostimulation Light OFF: 0.18 ± 0.05 Hz; Light ON: 0.23 ± 0.06; post-photostimulation Light OFF: 0.23 ± 0.06; one-way repeated measures ANOVA, $F$ (2, 12) = 3.4, Fig. 2d, f).

**Immediate and delayed photostimulation of PV+ interneurons**. We next asked whether the anti-seizure effect of photostimulation of PV+ interneurons depended on the timing of light delivery. To address this question, we varied the delay from the start of the seizure to the light pulse in our closed-loop photostimulation (Fig. 3a). Delays were randomised, and interspersed with 0 s delay trials, in every experiment to avoid any cumulative effect of laser activation. Furthermore, the light pulse was only triggered on alternate trials, so that the length of each event accompanied by photostimulation could be compared to

the length of the previous event, which acted as a control (Supplementary Fig. 2 and Fig. 3b). Consistent with the results shown in Fig. 2d and e, photostimulation of ChR2-expressing interneurons immediately upon seizure detection led to a 34.5 ± 7.7% reduction in seizure duration (Fig. 3b, 86 trials in eight mice, $p$ = 0.003, two-tailed paired $t$-test). This effect was independent of the length of the interval between control and photostimulated events suggesting that slow post-seizure recovery processes did not contribute (Supplementary Fig. 3). The seizure-suppressing action, however, was not observed when photostimulation was delayed by up to 2 s. Instead, light pulses delivered with a delay greater than 2 s (median 3.38 s, mean 3.63 ± 0.26 s) markedly prolonged seizures by 35.1 ± 5.8% (Fig. 3b; 20 trials in five mice, $p$ = 0.004, two-tailed paired $t$-test). Average durations of control non-stimulated seizures for each experimental condition are provided in Table 1 (see also Fig. 3b scatter plots for a systematic comparison of pairs of control seizures and subsequent 'stimulated' seizures used to trigger laser activation, for each delay). To verify that the prolongation of seizure duration by delayed photostimulation was not a result of a bias introduced by exclusion of shorter seizures from the ECoG analysis, we reshuffled the raw data by randomly assigning longer (>2.5 s) ictal events to the experimental group disregarding the information about the light pulse (pseudo-stimulated group). As in the case of the actual photostimulation, the preceding seizure was used for comparison. In contrast to the light-stimulated experimental group, no difference was found between seizure durations in the randomised group. The shuffled pseudo-stimulated and real dataset were

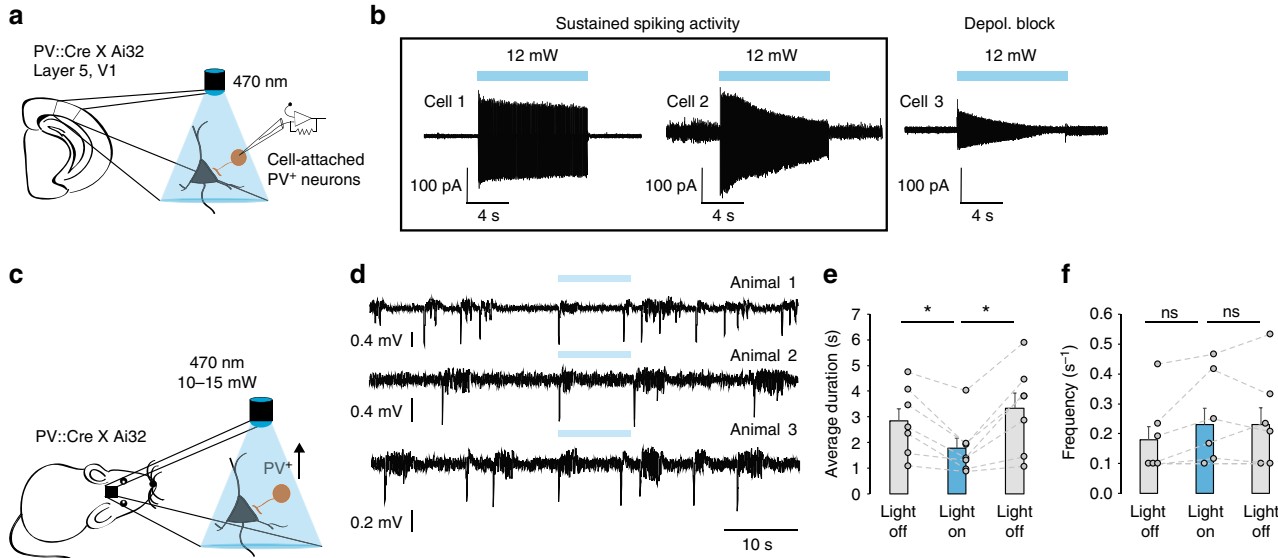

**Fig. 2** Photo-depolarization of ChR2-expressing PV+ interneurons reduces the duration of seizures. **a** Schematic of in vitro experiments and sample traces of cell-attached recordings from V1 layer 5 PV+ interneurons in acute brain slices. **b** Cell-attached recordings showed that most PV+ neurons are able to sustain firing during the 10 s-long light pulse (cells 1 and 2). Some cells (cell 3) show reduced firing towards the end of the light pulse. **c** In vivo experimental schematic. **d** ECoG traces showing epileptiform activity before, during and after several rounds of photostimulation of ChR2-expressing PV+ interneurons. Ictal discharges have reduced duration during optogenetic stimulation, but rapidly recover as soon as the light is switched off. **e** Average duration of individual ictal discharges during the 10 s periods before, during and after light stimulation ($n = 7$ mice, $*p < 0.05$, error bars represent s.e.m., one-way repeated measures ANOVA post hoc Bonferroni test). **f** Average number of ictal events during the 10 s periods before, during and after light stimulation ($n = 7$ mice, error bars represent s.e.m., one-way repeated measures ANOVA post hoc Bonferroni test). Source data are provided as a Source Data file

significantly different (normalised seizure duration in the randomised group: $99.39 \pm 5.05\%$; 57 trials in five mice, $p = 0.034$ for paired comparison, and 82 trials in nine mice, $p = 0.001$ for unpaired comparison with the photostimulated group; Fig. 3c). Finally, we verified that the reduction in seizure duration with immediate photostimulation was not an artefactual result of data selection by restricting analysis to seizures longer than 2.5 s: this still showed a robust anti-seizure effect of PV+ activation (Fig. 3d, e).

To ensure that the pro-ictal effect of delayed photostimulation was not due to a non-specific light-induced action during seizures or an unaccounted experimental bias, we tested if photoinhibition of PV+ interneurons had a different action. We transduced PV+ interneurons with the inhibitory opsin Arch3.0 using an AAV2/5 viral vector and confirmed the inhibitory action of photostimulation in vitro. The GABA$_A$ receptor-mediated synaptic drive to principal neurons in slices challenged with an aCSF designed to increase spontaneous activity (nominally 0 Mg$^{2+}$, 5 mM K$^+$) was profoundly suppressed (Supplementary Fig. 4a), and whole-cell recordings from Arch3.0-expressing PV+ interneurons displayed sustained hyperpolarization during photostimulation (Supplementary Fig. 4b). We then repeated closed-loop experiments, photo-hyperpolarizing Arch3.0-expressing cortical PV+ interneurons.

Unexpectedly, photo-induced activation of Arch3.0 with no or small (up to 2 s) delay had no detectable impact on seizure duration (Supplementary Fig. 5). However, this finding is consistent with the view that the inhibitory restraint has already failed to contain runaway excitation in the network by the time a seizure has been detected[2–4]. In contrast, when the laser was switched on with more than 2 s delay (median 3.05 s, mean $3.88 \pm 0.34$ s), light-induced hyperpolarization of PV+ interneurons significantly reduced the seizure duration by $29.1 \pm 4.7\%$ (Supplementary Fig. 5a–c, 31 trials in six mice, $p = 0.003$, two-

tailed paired $t$-test). Thus, light-induced hyperpolarization of these cells later in the seizure produces the opposite effect to photodepolarization—shortening rather than prolonging seizures. This anti-ictal effect was absent in a randomised control analysis group (normalised duration in the randomised group: $100.8 \pm 2.95\%$; 45 trials in six mice, $p = 0.023$ for paired, and 75 trials in nine mice, $p = 0.0003$ for unpaired comparison with the photostimulated group; Supplementary Fig. 5d).

Thus, delayed photo-depolarization or photo-hyperpolarization of PV+ interneurons by more than 2 s leads to paradoxical effects on in vivo seizure duration in the focal pilocarpine model. Depolarization, normally expected to enhance GABAergic inhibition, prolongs seizures, whilst hyperpolarization, expected to suppress inhibition, shortens them (Supplementary Fig. 5e). This strongly suggests that the functional role of these cells changes rapidly during sustained pathological network activity.

We asked whether the pro-ictal effect of optogenetic stimulation of interneurons is cell-type specific. We targeted expression of ChR2 to another well-defined, non-overlapping class of interneurons, somatostatin-positive (SOM+) cells. *SOM::Cre* and *Ai32* mouse lines were crossed to obtain progeny with conditional ChR2 expression (Supplementary Fig. 6a). Photostimulation initiated immediately upon seizure onset suppressed seizures, albeit less than when PV+ interneurons were photoactivated (normalised duration at 0 s delay: $78.2 \pm 4.18\%$, 155 trials in 13 mice, $p = 0.001$, two-tailed paired $t$-test, Supplementary Fig. 6b, c). Photostimulation delivered with 0.5 and 2 s delays after the onset of ictal discharges was similarly effective in suppressing seizures. With further delay to photostimulation of SOM+ interneurons the shortening of epileptic discharges was no longer seen (normalized duration of seizures photostimulated with >2 s delay: $95.92 \pm 5.77\%$, 36 trials in 5 mice, $p = 0.658$, two-tailed paired $t$-test). These results suggest that PV+ and SOM+ interneurons have distinct effects in cortical seizure perpetuation:

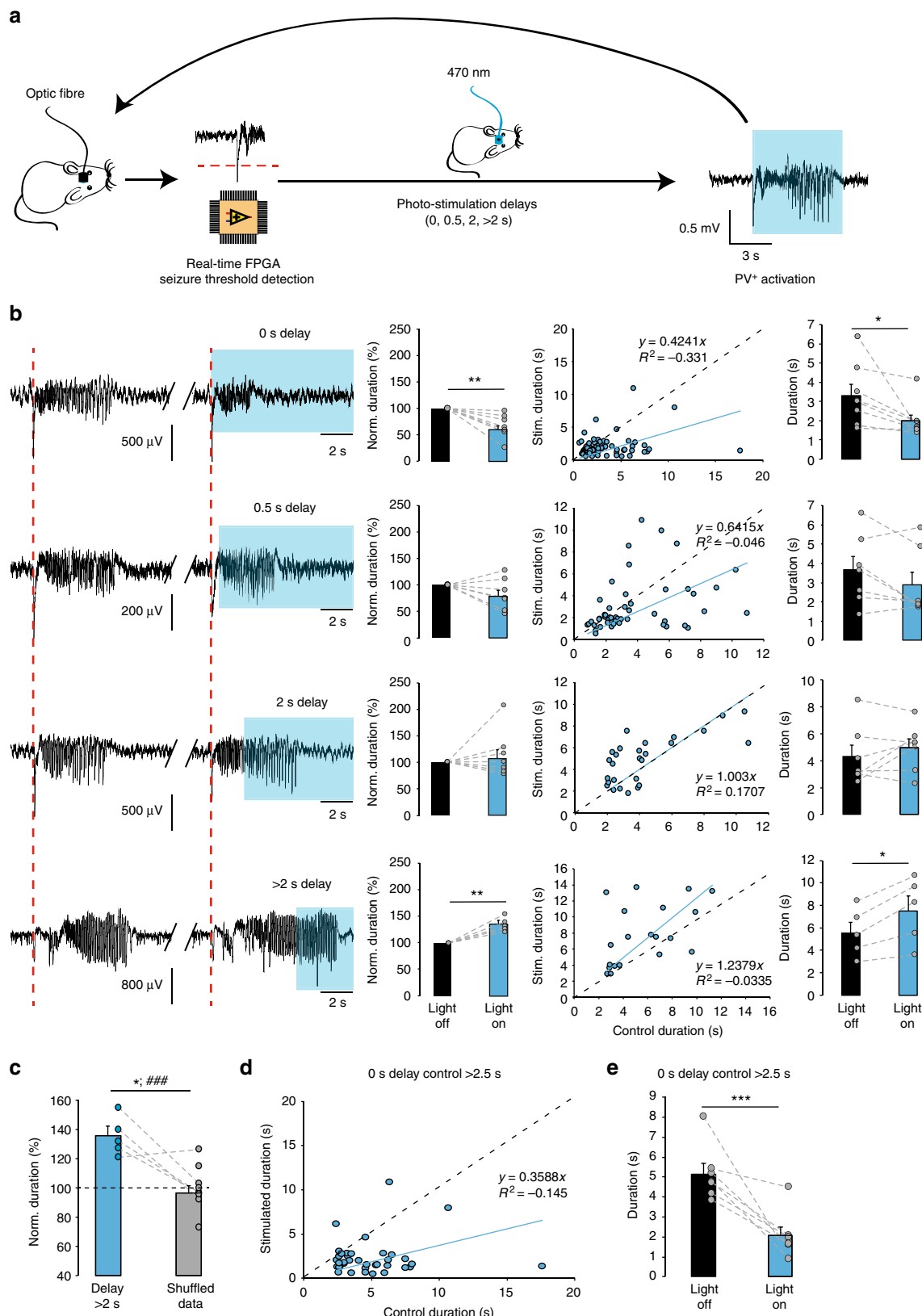

while depolarization of SOM+ interneurons becomes ineffective in reducing seizure duration after 2 s of ictal activity, PV+ interneurons rapidly switch their action and begin to promote electrographic seizures.

**KCC2 overexpression prevents pro-ictal action of PV+ cells.** Activity-induced intracellular chloride accumulation and a consequent depolarizing shift of the $GABA_A$ reversal potential ($E_{GABA}$) have been suggested to underlie the generation of

**Fig. 3** Dual effect of photo-depolarization of ChR2-expressing PV+ interneurons on the duration of seizures. **a** A closed-loop system was used to detect the onset of seizures and to vary the delay of optogenetic stimulation of ChR2-expressing PV+ neurons during pathological ECoG activity. FPGA field-programmable gate array. **b** Effects of photostimulation delivered at varying delays (increasing from top to bottom). Sample traces (left) illustrate pairs of consecutive electrographic seizures without and with photostimulation (blue rectangles indicate photostimulation). Intervening periods between seizures are omitted. The bar charts (middle and right) show a switch from light-induced seizure suppression with 0 and 0.5 s delays to seizure prolongation with >2 s delay (middle - normalised duration, right - absolute duration, $n = 8, 7, 7$ and 5 mice for 0, 0.5, 2 and >2 s delays respectively). In the scatter plots, each data point represents a pair of consecutive seizures, allowing a comparison of the duration of the discharge that was used to trigger the laser to that of the preceding control discharge. A linear regression line (blue) below the line of identity (dashed black) indicates that photo-depolarization reduced the seizure duration, whilst a line above the line of identity indicates prolongation of the seizure duration. **c** Normalised duration of ictal discharges during delayed photostimulation compared to randomized reshuffled data (*$p < 0.05$ paired $t$-test, $n = 5$; ### $p < 0.001$, unpaired $t$-test, $n = 9$, includes data from additional animals in randomized group). Error bars represent s.e.m. **d** Analysis of a subset of 0 s delay trials shown in **b** where the control seizure was longer than 2.5 s. The anti-seizure effect of immediate PV+ interneuron photostimulation persists, arguing against selection bias. **e** Summary of seizure durations in **d**. Statistics in **b** and **e**, paired $t$-test. *$p < 0.05$, **$p < 0.01$, ***$p < 0.001$; error bars represent s.e.m. Source data are provided as a Source Data file

**Table 1 Mean duration of baseline ictal discharges in experiments with different delays in different experimental groups**

| Delay | PV-ChR2 | PV-ChR2/CaMKII-GFP | PV-ChR2/CaMKII-KCC2 |
|---|---|---|---|
| 0 s | 3.31 ± 0.56 | 2.09 ± 0.38 ns | 2.28 ± 0.23 ns |
| 0.5 s | 3.66 ± 0.69 | 2.09 ± 0.48 ns | 3.0 ± 0.2 ns |
| 2 s | 4.32 ± 0.81 | 4.31 ± 0.63 ns | 4.78 ± 0.67 ns |
| >2 s | 5.52 ± 0.96 | 5.45 ± 0.49 ns | 5.45 ± 0.47 ns |

Data shown as mean ± s.e.m.
*ns* non-significant (compared to PV-ChR2 group; one-way ANOVA followed by a post hoc Bonferroni correction for multiple comparisons)

seizure afterdischarges in response to brief activation of interneurons or GABA uncaging in vitro[19]. However, chloride regulation critically depends on energy metabolism and could easily be disrupted in cultured neurons and following the trauma of acute brain slice preparation[22], and so the degree and impact of seizure-associated changes in $E_{GABA}$ in vivo remain uncertain. The capacity of adult neurons to extrude chloride is mainly determined by the neuron-specific potassium-chloride co-transporter KCC2. Therefore, to establish whether regulation of intracellular chloride homeostasis is involved in the rapid conversion of the anti-ictal action of PV+ photostimulation into a pro-ictal one, we overexpressed KCC2 in cortical principal neurons. To drive the expression of the transgene in the target neurons, we designed a lentiviral vector containing a KCC2 construct[23] under a human CaMKII (hCaMKII) promoter (hCAMKII-KCC2-IRES-TdTomato), which was injected into the mouse visual cortex. To validate the functional effects of the transgene in the transduced neurons, we first compared the depression of GABA$_A$ receptor-mediated currents during electrical stimulation in control and neighbouring KCC2 overexpressing layer 5 pyramidal neurons from the same slices in the presence of the ionotropic glutamate and GABA$_B$ receptor blockers. GABA$_A$ currents were evoked by trains of electrical stimuli (10 pulses at 10 Hz). Cells with transgene expression displayed a reduced depression of GABAergic neurotransmission during stimulation trains compared to non-transduced neurons (two-way ANOVA, $F (1, 11) = 8.035$, $p = 0.016$, $n = 7$ transduced neurons vs. $n = 6$ non-transduced neurons, Fig. 4a–c). We next tested the efficacy of the chloride extrusion mechanism in control and transgene-expressing pyramidal neurons. To do this we compared pyramidal neurons transduced with the GFP-tagged light-sensitive chloride pump halorhodopsin (Halo-GFP) alone and those co-transfected with the KCC2-carrying construct and Halo-GFP. Neurons were loaded with chloride by

optogenetic activation of halorhodopsin and simultaneous membrane depolarisation (see Fig. 4d, e, and Methods). GABA$_A$-mediated currents were recorded in response to brief GABA puffs before and after chloride loading (Fig. 4d, e). While the photocurrents used to load chloride in both groups were similar (peak amplitude Halo: 135 ± 9.57 pA vs Halo + KCC2: 118.6 ± 10.33 pA, unpaired $t$-test, $p = 0.26$, $n = 7$ per group, Fig. 4f), neurons co-expressing Halo and KCC2 displayed a three-fold faster recovery from chloride loading compared to neurons expressing Halo alone (recovery time constant Halo: 46.5 ± 12.7 s vs Halo + KCC2: 16.6 ± 4.0 s, unpaired $t$-test, $p = 0.046$, $n = 7$ per group, Fig. 4g–i).

We then expressed the KCC2 construct in the cortex of PV:: Cre × Ai32 mice (Fig. 5a, b). This did not change the effect of continuous optogenetic activation of ChR2-expressing PV+ interneurons on ictal activity (Fig. 5c, d, c.f. Fig. 2d, e). In agreement with the earlier experiments, closed-loop photostimulation of PV+ cells immediately after seizure onset and at a 0.5 s delay consistently reduced the duration of epileptiform discharges (0 s delay: 72.4 ± 4.3%, 94 trials in 10 mice, $p = 0.0001$; 0.5 s delay: 84.6 ± 5.5%, 62 trials in seven mice, $p = 0.031$, two-tailed paired $t$-test). However, the seizure prolongation by delayed photo-depolarization of PV+ cells was completely lost (average seizure duration photostimulated with >2 s delay: 99.4 ± 5.5 % of control seizure duration, 50 trials in eight mice; Fig. 5e). In contrast, consistent with the earlier experiments, control animals transduced with a lentivirus expressing GFP under the same CaMKII promoter exhibited a reduction of seizure duration with immediate photo-depolarization of PV+ interneurons (74.1 ± 5.8%, eight mice), and seizure prolongation with delayed photo-depolarization (150.8 ± 13.5%, 29 trials in six mice; Fig. 5f and Supplementary Fig. 7). The average durations of control, non-stimulated seizures for each experimental condition were similar in PV-ChR2, PV-ChR2/CaMKII-KCC2 and PV-ChR2/CaMKII-GFP mice (Table 1). These results imply that KCC2 activity does not interfere with seizure initiation, but affects seizure maintenance when intracellular chloride starts to accumulate.

We conclude that limited ability of principal neurons to resist chloride loading during excessive network activity contributes to the seizure-promoting action of PV+ interneurons that develops within seconds of seizure onset. Increasing the neuronal capacity to extrude chloride by overexpressing KCC2, therefore, can prevent interneurons from facilitating seizures.

## Discussion
Interneurons constitute a highly heterogeneous population of cells characterized by diverse firing properties, projection targets

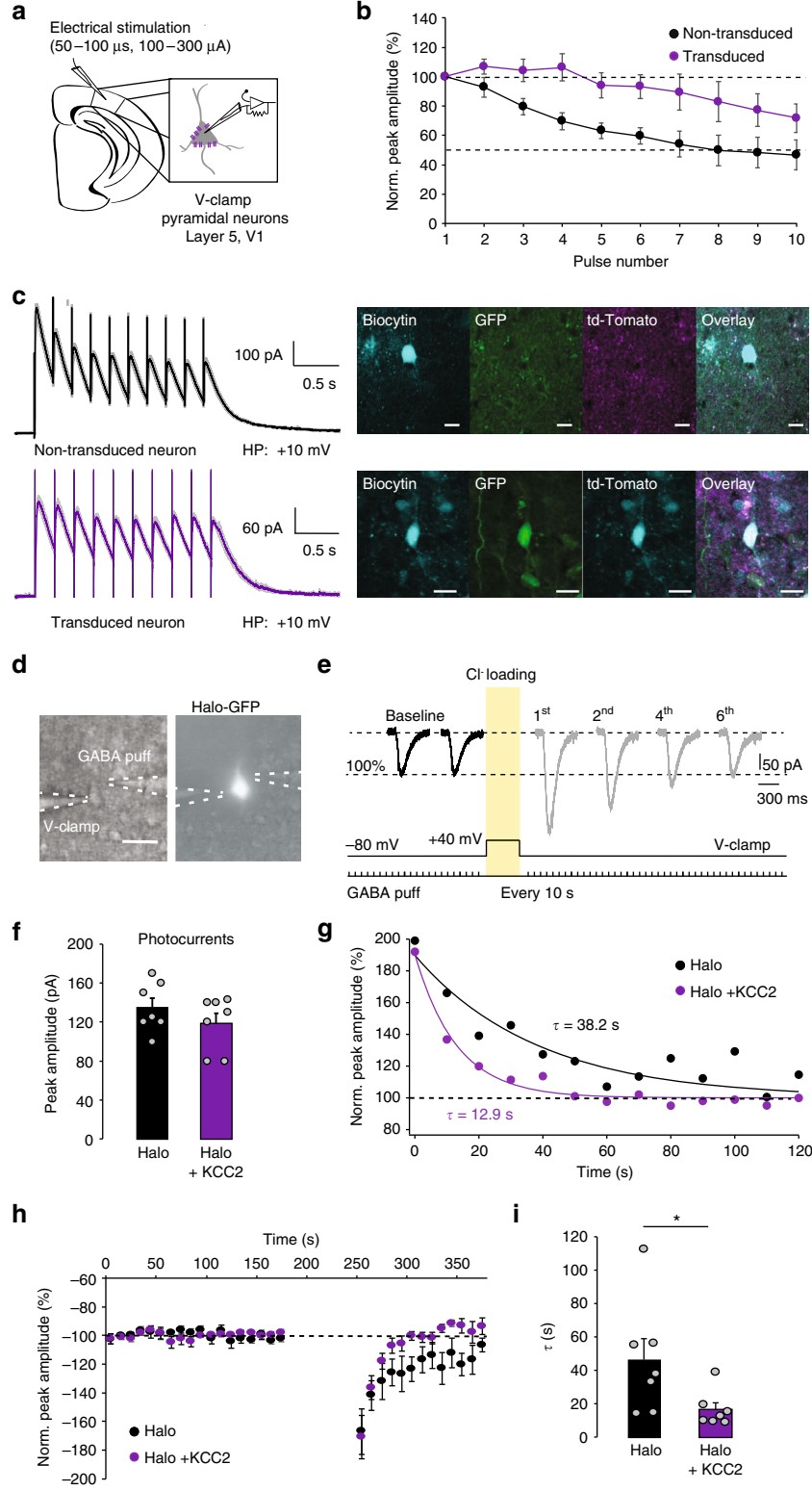

and roles in regulating network dynamics[24,25]. With a few notable exceptions[26–30] the firing of individual interneuronal subtypes at different stages of epileptiform activity is largely unknown, as are their relative contributions to seizure initiation and termination. Using a model of acute focal cortical seizures in awake behaving mice, we have shown that optogenetic depolarization of perisomatic-targeting PV+ interneurons can have opposite

effects on seizures. Photostimulation of PV+ cells triggered upon seizure onset decreases the duration of ictal discharges. However, photostimulation delayed by 2 s or more prolongs electrographic seizures. We have further established that overexpression of KCC2 in pyramidal neurons located close to the seizure initiation site prevents the pro-ictal effect of delayed PV+ photostimulation. This provides strong evidence that the capacity of neurons

**Fig. 4** Overexpression of the KCC2 co-transporter in cortical principal neurons. **a** Experimental schematic. **b** Normalized peak amplitudes of consecutive evoked GABA$_A$ receptor-mediated currents recorded in the transgene-expressing (purple; $n = 7$) and neighbouring non-transduced (black; $n = 6$) layer 5 V1 pyramidal neurons in acute brain slices in response to 10 electrical pulses delivered at 10 Hz. **c** Sample traces show averages of 10 trials from non-transduced and transduced neuron (95% confidence intervals displayed in grey) as well as their recovered immunofluorescence, scale bar: 10 μm. HP holding potential, GFP green fluorescent protein, V-Clamp voltage-clamp. **d**, **e** Experimental protocol. The light-sensitive chloride pump halorhodopsin was activated by photostimulation for 1 min while a pyramidal neuron was held at $+40$ mV to load it with chloride. Scale bar: 20 μm. Raw traces in baseline and after Cl$^-$ loading. **f** Photostimulation to activate halorhodopsin was adjusted to induce photocurrents of a similar peak amplitude in control neurons and in neurons overexpressing KCC2. **g** Time course of GABA$_A$ receptor-mediated currents recovery from Cl$^-$ loading in layer 5 V1 pyramidal neuron expressing halorhodopsin alone (black) and in a cell co-expressing halorhodopsin and KCC2 (purple). Note that the recovery time is three times faster in the KCC2-expressing neuron. **h** Averaged normalized peak amplitudes of GABA$_A$ receptor-mediated currents evoked by 10 ms GABA puffs (100 μM) in layer 5 V1 pyramidal neurons expressing halorhodopsin alone (black, $n = 7$) and in cells co-expressing halorhodopsin and KCC2 (purple; $n = 7$). **i** Recovery time constants of GABA$_A$ receptor-mediated currents calculated from single exponential fitting (from the first post-loading measure to 2 min recovery). Neurons overexpressing KCC2 show significantly faster recovery than control neurons. Unpaired t-test, *$p < 0.05$, error bars represent the s.e.m. Source data are provided as a Source Data file

to extrude chloride is overwhelmed during seizures and that a collapse of voltage inhibition contributes to perpetuating seizures.

The evidence for a causal role of impaired chloride extrusion reported here complements evidence from human epilepsy and animal models. Downregulation of KCC2 and/or upregulation of the chloride importer NKCC1 have been reported in human focal epilepsy, and loss-of-function KCC2 mutations cause severe early-onset epilepsy[6,31–36]. Conversely, potentiating KCC2 activity limits seizure severity[37]. Failure of hyperpolarizing inhibition has recently also been proposed to underlie the aggravation of kindling-induced generalized seizures by optogenetic stimulation of subicular interneurons in mice[38].

In addition to a chronic deficit in GABA$_A$ receptor-mediated inhibition, its loss, and even a switch to excitation, may occur rapidly as a consequence of activity-dependent chloride loading[39,40]. The latter mechanism has long been postulated to exacerbate seizures, and has gained experimental support from recent in vitro findings that brief activation of GABA$_A$ receptors during electrographic seizures can initiate an after-discharge[19] and that a rapid intracellular chloride increase can be reliably detected in vivo within the first second of a seizure[41]. Chloride accumulation evolves with fast LFP oscillations, and reverses — at about the same rate, if not slower — after the termination of seizure activity[42]. The amount of accumulated intracellular chloride correlates with the length of the seizure[41]. Full recovery of the resting trans-membrane chloride gradient could take several tens of seconds after a long seizure, similar to the situation observed in our in vitro experiments where photo-stimulation of halorhodopsin for 1 min was used to load chloride into pyramidal neurons. We note, however, that the recovery rates estimated with the invasive electrophysiological techniques in vitro may not accurately reflect the dynamics of the ionic changes in vivo[43]. While slow recovery of the intracellular chloride concentration after a seizure theoretically might impact on the next ictal event, in our experiments, photostimulation of PV+ cells at the onset of seizures consistently had an anti-seizure effect, suggesting that the inter-ictal intervals in our model were sufficient to restore the inhibitory influence of these interneurons.

Our study provides, to our knowledge, the first in vivo demonstration that PV+ interneurons can switch their action during a seizure and prolong on-going ictal activity in non-anaesthetized freely moving animals.

The time point at which PV+ interneurons lose their ability to curtail ictal events depends on multiple factors, including the efficiency of KCC2-mediated chloride extrusion and the rate of GABA$_A$ receptor-mediated neurotransmission. This is likely to vary between human epilepsies and different animal models, possibly explaining why, in a recent study, optogenetic

stimulation of PV+ interneurons in anaesthetized mice was only found to curtail seizures induced by topical application of 4-AP onto the neocortex[44]. Alternatively, a pro-seizure action of PV+ cells might have been overlooked in that study, since the delay from seizure onset to the laser pulse was not controlled (2.9 s on average), and the effects of photostimulation were excluded from the analysis if delays were longer than 5 s.

Surprisingly, immediate photo-hyperpolarization of PV+ interneurons at seizure onset had no effect in the present study. This could be explained by postulating that PV+ interneurons already failed to contain seizure spread and their suppression therefore had little additional impact. However, other possible explanations for this unexpected result are worth considering. For example, the experimental design was not exactly comparable, because the hyperpolarizing opsin was expressed with a viral vector, whilst ChR2 was expressed by crossing PV::Cre and Ai32 mice. Furthermore, we cannot exclude the possibility that early in the seizure, our photostimulation mainly hyperpolarized the axons of those interneurons that were still outside the pilocarpine-activated circuit; and that these cells might only become activated later in the seizure as pathological activity spread.

KCC2 overexpression using a very similar construct has previously been demonstrated to convert GABA from depolarizing to hyperpolarizing in neonatal rats[45]. Viral overexpression of KCC2 in mature healthy neurons would not, however, be expected to have a major effect on hyperpolarising E$_{GABA}$[46], a feature that makes this intervention attractive from a translational perspective[47]. Further research is required to fully establish the therapeutic potential of KCC2 gene manipulation for epilepsy treatment. A potential concern is that increased co-transport of chloride and potassium ions may exacerbate activity-induced extracellular potassium transients and so facilitate seizure initiation and/or maintenance[48,49]. Existing evidence, however, suggests otherwise—abolishing phospho-dependent inactivation of KCC2, thus increasing its activity in the knock-in mice, delayed seizure onset in vitro and limited seizure severity in an in vivo kainate model[37]. Here, we focused on the contribution of KCC2 to interneuron function during seizures and did not system-atically explore the impact of KCC2 overexpression on seizure susceptibility. We did not observe any effect of KCC2 over-expression on the ability of immediate photoactivation of PV+ interneurons to curtail seizures, suggesting that KCC2 activity was minimal at the start of a seizure and was not contributing to seizure initiation in our model. We also note in this respect, that the current study concentrated on acute seizures, while various changes associated with chronic condition, including cell type-specific loss of inhibitory interneurons, may contribute to the

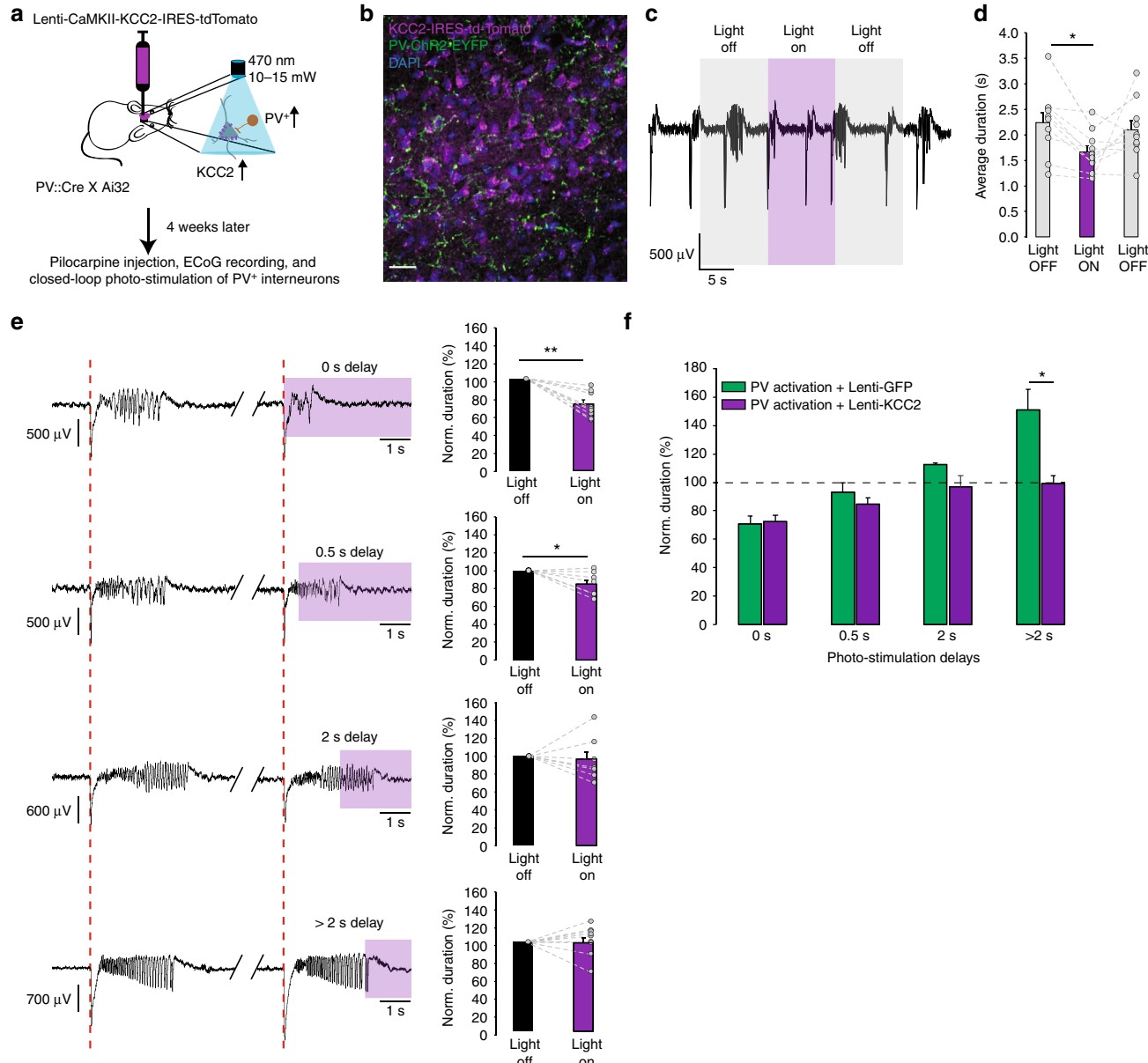

**Fig. 5** KCC2 overexpression in principal neurons prevents the pro-ictal action of PV+ photoactivation during seizures. **a** Experimental schematic. **b** Immunofluorescence of the PV-ChR2-EYFP (green), the nuclei staining, DAPI (blue), and the KCC2-IRES-tdTomato (magenta) in an overlaid image. Scale bar 25 μm. **c** ECoG recordings of ictal discharges before, during and after photoactivation of ChR2-expressing PV+ interneurons in animals that overexpress KCC2 in the cortical principal neurons at the site of pilocarpine injection. **d** Average duration of ictal discharges during the 10 s baseline (grey) and photostimulation (purple) periods. KCC2 overexpression does not affect the reduction of seizure duration by PV+ photoactivation (c.f. Fig. 2; $n = 8$ mice, one-way repeated measures ANOVA post hoc Bonferroni correction, *$p < 0.05$, error bars represent the s.e.m.). **e** KCC2 overexpression abolishes prolongation of seizures by delayed (>2 s) PV+ photoactivation ($n = 10, 7, 7$ and 8 mice for the delays of 0, 0.5, 2 and >2 s, paired $t$-test). Sample traces illustrate pairs of consecutive seizures without and with photostimulation (intervening periods between seizures are omitted; purple rectangles indicate photostimulation). **f** Normalised duration of ictal discharges by photostimulation of PV+ neurons in mice with (purple) KCC2 overexpression or GFP expression (green). Note that KCC2 overexpression has no significant effect on the seizure suppression when photostimulation of PV+ interneurons is delivered immediately upon seizure detection. *$p < 0.05$, **$p < 0.01$, ***$p < 0.001$; error bars represent s.e.m. Source data are provided as a Source Data file

mechanisms of ictogenesis in various epilepsies. Indeed, it has been suggested that reduced KCC2 expression, often associated with chronic epilepsy, may represent an adaptive anti-epileptic mechanism[49].

Intracellular chloride regulation has been shown to differ between axons, somata and dendrites[50–53]. While the chloride gradient might be expected to collapse more rapidly in thin dendrites compared to the somatic region, we found that delayed optogenetic stimulation of dendritic-targeting SOM+ interneurons did not facilitate seizures. These results are consistent with a previous report that seizure-induced $E_{GABA}$ shifts are more prominent in the soma than in dendrites of pyramidal neurons[19]. Our findings are also in accord with the recent demonstration that photoactivation of PV+, but not SOM+ interneurons aggravates seizures in the kindled hippocampus with perturbed KCC2 expression[38].

Contrasting actions of PV+ and SOM+ interneurons may also reflect a difference in either the extent of their activation during seizures, the density of GABAergic synapses between perisomatic and dendritic regions[54], or a non-uniform KCC2 distribution along the neuronal membrane. Indeed, the axon initial segment is potentially vulnerable to a shift in $E_{GABA}$ because of its small size and dense innervation by PV+ chandelier cells, and has a critical role in action potential initiation. Overexpressing KCC2 specifically in the axonal, somatic or dendritic compartment of pyramidal neurons and testing the effects of optogenetic activation of different interneurons would help to address this question. Irrespective of the exact underlying mechanism, our results suggest that during epileptiform discharges two simultaneous GABA_A receptor-mediated inputs can have very different functional impacts on neuronal excitability depending on their spatial location. This activity-dependent dichotomy between perisomatic-targeting and dendritic targeting GABAergic connections further adds to the complexity of the roles played by distinct populations of interneurons in modulating the output of pyramidal cells[55,56]. In addition, in chronic epilepsy, inhibitory control of network excitability may be impaired because of cell type-specific loss of interneurons. It remains to be established whether reduced dendritic, but maintained perisomatic GABAergic inputs, as observed in animal studies and in humans[57,58], contribute to network instability and generation of recurrent spontaneous seizures.

The results of the KCC2 overexpression experiments in this study strongly suggest a role for a seizure-induced $E_{GABA}$ shift in rapid weakening of the inhibitory action of PV+ interneurons during ictal activity. This, however, may not be the only mechanism responsible for the dual effect of PV+ photostimulation on seizure duration and/or of the anti-seizure effect achieved by photoinhibition of PV+ interneurons. Alternatively, activity-induced depolarization block of action potential firing of PV+ interneurons may be facilitated by the delayed optogenetic depolarization of these cells[29,30,59]. The two mechanisms are not mutually exclusive and may both contribute to sustaining seizures.

In conclusion, the dual effect of optogenetic stimulation of PV+ cells discovered in our study explains existing controversies regarding the role of this group of interneurons in seizure initiation and maintenance.

## Methods

**Animals**. Animal experiments were conducted in accordance with the United Kingdom Animal (Scientific Procedures) Act 1986, and approved by the Home Office (license PPL70-7684). *PV::Cre* (*B6;129P2-Pvalb<sup>tm1(cre)Arbr</sup>*/J, RRID: IMSR_JAX:008069) and *SOM::Cre* (*B6N.Cg-Ssttm2.1(cre)Zjh*/J, RRID: IMSR_JAX:013044) mice were either bred with the Ai32 mouse line (*B6;129S-Gt (ROSA)26Sor<sup>tm32(CAG-COP4*H134R/EYFP)Hze</sup>*/J, RRID: IMSR_JAX:012569) for *Cre*-inducible conditional expression of EYFP-tagged ChR2 in target interneurons, or locally transfected with an adeno-associated viral vector carrying Arch3.0 construct (*AAV-dio-Arch3.0-EYFP*, serotype 2/5, from UNC vector core). Animals were housed individually on 12 h/12 h dark/light cycle, and food and water were given ad libitum.

**Surgery and viral injection**. Animals of both sexes (P60–P120) were anaesthetized with isoflurane and placed in a stereotaxic frame (David Kopf Instruments Ltd., USA). After removal of the skin and cleaning of the skull, a hole was drilled above the right primary visual cortex to implant a cannula (Bilaney Consultants Ltd.) and place the ECoG electrodes (coordinates: Antero-Posterior: −2.8 mm from the bregma; Lateral: 2.4 mm). Wireless mouse 512 Hz single channel ECoG transmitters (A3028B-CC, Open Source Instruments Inc., USA) were implanted subcutaneously with the recording electrode positioned on the dura above the injection site and the reference electrode placed on the contralateral side. The cannula and the ECoG electrodes were tightly fixed using cyanoacrylate glue and dental cement. Animals were injected with buprenorphine (0.02 mg/kg) and metacam (0.1 mg/kg) at the beginning of the surgery and with at least 0.5 ml of saline at the end. Mice were checked daily and allowed to recover for at least a week before experiments commenced.

To overexpress GFP or KCC2 in principal cortical neurons, we produced a lentiviral vector (*hCAMKII-GFP or hCAMKII-KCC2-IRES-TdTomato*, VectorBuilder, USA) containing a KCC2 construct (Addgene, MA, USA, plasmid #61404)[23] under the *hCaMKII* promoter[60]. For Arch3.0, GFP and KCC2 expression, viral injections were performed just prior to implantation of the cannula at two locations (0.55 and 0.35 mm below the pia) using a microinjection pump (WPI Ltd., USA), 5 μl Hamilton syringe (Esslab Ltd., UK), and a 33 gauge needle (Esslab Ltd., UK) (injection volume per site: 200 nl, rate: 100 nl/min). Optogenetic experiments were performed 3–4 weeks after viral injection allowing sufficient time for transgene expression to reach a stable level.

**In vivo seizures and closed-loop optogenetic stimulation**. Animals were briefly sedated with isoflurane and placed in a stereotaxic frame. Highly concentrated pilocarpine in sterile saline (3M) was injected 0.5 mm below the pia via the implanted cannula guide (Bilaney Consultants Ltd., UK) positioned over the primary visual cortex. Approximately 200–400 nl were injected at a rate of 100 nl/min. We waited 10 min before withdrawing the needle to avoid backflow of the solution.

After injection, the needle in the cannula was replaced by an optic fibre (0.22 NA, 200 μm Ø, Multimode 190–1200 nm, Thorlabs Inc., UK) inserted into the brain at the same depth. The animal was then transferred to a home cage placed in the experimental setup for ECoG recording, and data acquired at 512 Hz using the Neuroarchiver tool (Open Source Instruments Inc., USA). Epileptiform activity typically appeared 10 min after the animal recovered from the pilocarpine injection performed under anaesthesia. In rare cases where epileptic discharges evolved into continuous activity (see Fig. 1) and persisted for more than an hour, intra-peritoneal injection of diazepam (10 mg/kg) was used to terminate the experiment. Up to three injections of pilocarpine on three separate days were performed to collect data from each experimental animal. To monitor the motor behaviour and the ECoG simultaneously during pilocarpine-induced seizures, animals implanted with wireless transmitters were placed in a transparent chamber surrounded by three IP cameras (MicroSeven). The video stream was synchronised with ECoG using the MicroSeven software.

Optogenetic stimulation was performed through the optic fibre connected to the laser light source (wavelength: 470 nm for photoactivation of ChR2 or 570 nm for photoactivation of Arch3.0; power intensity at the tip of the fibre: 10–15 mW and 5–10 mW respectively). The ECoG signal was monitored on-line and the laser was automatically triggered with various delays in closed-loop by a field-programmable gate array (FPGA) CRIO-9076 integrated controller (National Instruments, USA) upon seizure detection. The delays of photostimulation were randomized. The seizure detection threshold was adjusted at the start of each experiment depending on the amplitude of the ECoG signal and the magnitude of noise. The correct timing of light stimulations was then confirmed during the off-line analysis of the data.

**Automated network state classifier and ECoG analysis**. Pilocarpine-induced epileptiform activity generally fell into the three distinct categories (Fig. 1; S1—interictal, S2—ictal-like, S3—continuous epileptiform discharges), with periods of quiescence (baseline). However, a significant proportion of recordings were ambiguous. In order to categorize network activity in an objective manner, and to exclude mixed states from analysis, a supervised machine learning approach was taken. Recordings were first min-max scaled to between 0 and 1, and for every 10 s, 20 features were extracted. Features from unambiguous periods were used to train a random forest classifier[21] and performance assessed on unseen test periods (>0.9 f1-score). The classifier was then used to objectively categorize activity proceeding photostimulation into one of the four states. To exclude mixed states classifier prediction probability was used: if the probability of the most likely state was less than twice the second most likely state, the period was deemed to be mixed. Manual inspection of the raw data identified in this manner verified this approach as appropriate. For 2D visualisation of features and their corresponding states, linear discriminant analysis was used to project the 20 features to 3 dimensions (Fig. 1c). Features used to discriminate between the different states were as follows: mean, standard deviation, kurtosis, skew, sum-abs-difference, number of peaks, number of valleys, mean valley, mean peak, average range between peak and valley, average wavelet power at 1, 5, 10, 15, 20, 30, 60, 90 Hz, number of baseline points, baseline points mean index difference.

To analyse the effects of photostimulation on the duration of seizures, each photostimulated discharge was compared to the immediately preceding non-stimulated one (Supplementary Figure 2b). However, in 27% of the trials with delays exceeding 2 s, control discharges were shorter than the delay itself. To avoid biasing the results towards photostimulation-induced prolongation of the activity, in these cases the nearest preceding discharge, which exceeded the duration of the delay was used as a control (Supplementary Figure 2c). The average duration of the photostimulation delay did not differ between the PV-ChR2 group and the other conditions (One way ANOVA, $F_{(4, 157)} = 2.199$, post hoc Bonferroni correction, PV-ChR2 average delay 3.64 ± 0.26 s vs. 3.88 ± 0.34 s for PV-Arch3.0, $p = 1.0$; vs. 3.98 ± 0.27 s for SOM-ChR2, $p = 1.0$; vs. 3.19 ± 0.11 s PV-ChR2/CaMKIIKCC2, $p = 1.0$, vs PV-ChR2/CaMKII-GFP 3.91 ± 0.36 s, $p = 1$, PV-ChR2/CaMKII-GFP vs PV-ChR2/CaMKIIKCC2 $p = 0.87$).

To generate the randomised data for the Fig. 3c and Supplementary Figure 5d, we re-analyzed ECoG recordings disregarding associated information about

photostimulation and only included seizures, the duration of which exceeded 2.5 s. Seizures were randomly assigned to the pseudo-stimulated group and their durations were compared to the durations of previous discharges (simulating experimental conditions). If the immediately preceding seizure length was less than 3 s, we searched for the closest qualifying event (Supplementary Figure 2).

**In vitro electrophysiology and optogenetic stimulation**. For in vitro electrophysiological recordings, animals (P60–P120) were sacrificed using an overdose of isoflurane. After decapitation brains were rapidly removed, and acute 350 µM-thick brain slices were prepared on a Leica VT1200S vibratome (Germany). The slicing was performed in ice-cold slicing solution that contained (in mM): 75 sucrose, 87 NaCl, 22 glucose, 2.5 KCl, 7 MgCl₂, 1.25 NaH₂PO₄, 0.5 CaCl₂ and 25 NaHCO₃, and was bubbled continuously with 95% O₂ + 5% CO₂ to yield a pH of 7.4 (315–330 mOsm). Slices were then placed in a sucrose-free aCSF solution (in mM): 119 NaCl, 2.5 KCl, 1.3 MgSO₄, 2.5 CaCl₂, 26.2 NaHCO₃, 1 NaH₂PO₄, 22 glucose, bubbled with 95% O₂ and 5% CO₂, incubated at 34 °C for 15 min, allowed to equilibrate to room temperature for another 15 min, and then kept in an interface chamber for at least 1 h at room temperature, before being transferred to a submerged recording chamber. Modified epileptogenic aCSF, which nominally contained 0 mM Mg²⁺ and 5 mM K⁺ was used in some experiments to facilitate spontaneous neuronal activity. All recordings were done at 32 °C.

Whole-cell patch-clamp recordings were performed from cortical and hippocampal interneurons and pyramidal cells visualized using an infrared differential contrast imaging system. Standard-walled borosilicate glass capillaries were used to fabricate recording electrodes with a resistance of 2.5–3.5 MΩ. The intracellular pipette solution for voltage-clamp recordings contained (in mM): 120 Cs-methanesulfonate, 10 HEPES, 0.2 EGTA, 8 NaCl, 0.2 MgCl₂, 2 Mg-ATP, 0.3 Na-GTP, 5 QX314-Br, 10 phosphocreatine, pH adjusted to 7.2 and osmolality adjusted to 296 mOsm. Series resistance was monitored throughout experiments using a −5 mV step command. Cells showing a series resistance > 20 MΩ, more than 20% change in series resistance, or unstable holding current were rejected. Current-clamp whole-cell recordings were performed using an intracellular solution containing (in mM): K-gluconate (145), NaCl (8), KOH-HEPES (10), EGTA (0.2), Mg-ATP (2), Na₃-GTP (0.3); pH 7.2; 290 mOsm. Cell-attached recordings of neuronal firing were done using patch pipettes (8–12 MΩ) filled with aCSF in the voltage-clamp mode with the voltage set so that no current was injected under baseline conditions.

For experiments shown in Fig. 4, the KCC2 construct was co-injected with a lentivirus carrying *hCaMKII-cop-GFP* since the IRES system on the *hCAMKII-KCC2-IRES-TdTomato* constructs did not allow direct visualisation of the transduced neurons by epifluorescence in non-fixed brain tissue. Using immunohistochemistry staining we found that at least 62.5 ± 4.2% (n = 3 slices) of GFP-expressing principal neurons were co-labelled with the RFP staining (labelling the tdTomato protein, Fig. 4c). Therefore, since the GFP-expressing group in these experiments might have had cells with no overexpression of KCC2, the difference between the control and KCC2-overexpressin neurons was likely to be underestimated. For experiments shown in Fig. 4d–i, the KCC2 construct was co-injected with a lentivirus carrying *hCaMKII-FCK-Halo-GFP* (Penn Vector Core, Addgene, MA, USA, plasmid #14750). AMPA and NMDA receptor blockers, NBQX (10 µM) and D-AP5 (50 µM) respectively, as well as the GABA_B antagonist, CGP55845 (1 µM), were bath-applied to isolated GABA_A currents evoked by either electrical stimulation or puff-application of GABA. In experiments with electrical stimulation, neurons were held at +10 mV and 10 pulses (duration of 50–100 µs) at 10 Hz were applied using a bipolar electrode at 100–300 µA current intensity. For puff-application, 100 µM GABA (dissolved in aCSF) was puffed through a borosilicate glass pipette (resistance 1–2 MΩ) at 3–10 PSI using a pressure injector (npi electronic GmBH). The pipette tip was placed 10–50 µm from the soma of a recorded neuron (Fig. 4d). GABA_A currents evoked by 10 ms puffs were recorded every 10 s while holding the neuron at −80 mV. After a stable baseline lasting at least 3 min, the chloride loading protocol was applied (Fig. 4e). Neurons were held at + 40 mV, and Halo was photostimulated continuously for 1 min (wavelength: 590 nm) while GABA puffs were interrupted. The recovery of GABA_A currents following the chloride loading was then monitored by holding the neuron at −80 mV. The light power was adjusted to produce photocurrents of similar magnitude in both Halo and KCC2 groups (Fig. 4f).

Recordings were obtained using a MultiClamp 700B amplifier (Axon Instruments, Foster City, CA, USA), filtered at 4 kHz and digitized at 10 kHz. Data acquisition and off-line analysis were performed using WinEDR 3.2.7 (University of Strathclyde, Glasgow, UK) and Clampfit 10.0 (Molecular Devices Corporation, USA) software.

Wide-field photostimulation of interneurons in acute brain slices was delivered through 20× water immersion objective (Olympus). Blue (wavelength 470 nm) or yellow (wavelength 570 nm) light was generated using pE-2 LED illumination system (CoolLED); light intensity at the surface of the slice was in the range of 5–12 mW.

**Immunohistochemistry**. For the KCC2 immunostaining, the brains were collected at the end of each experimental session. Animals were perfused with 4% paraformaldehyde (PFA, Santa Cruz), brains were extracted and left in PFA overnight before being washed 3–4 times with PBS sodium azide (0.02%), and stored at 4 °C

until being processed further. Brain slices (70 µm-thick) were then prepared using a Leica VT1200S vibratome (Germany) and first placed in 1 ml PBS 0.1 M, pH 7.4, in 12-well plates followed by 1 h incubation at room temperature in a blocking solution containing PBS 0.1 M, 0.5% Triton X100, 0.5% BSA and 3% donkey or goat serum. This step was followed by an overnight incubation in the primary polyclonal guinea pig anti-GFP antibody (Synaptic Systems Cat# 132 005 RRID: AB_11042617, 1:1000) and polyclonal rabbit anti-RFP antibody (Rockland Cat# 600-401-379 RRID: AB_2209751, 1:500) at 4 °C on a shaking plate (PBS 0.1M, 0.3% triton X100, 0.5% BSA). Slices were subsequently rinsed 3 times in PBS and incubated with secondary donkey anti-guinea antibody conjugated to Alexa Fluor 488, and donkey anti-rabbit conjugated to Alexa Fluor 568 (in PBS dilution 1:1000, Invitrogen) for 3 h at room temperature keep in the dark. For the biocytin filling, slices were also incubated with Alexa Fluor 405 Streptavidin (ThermoFisher scientific S32351, 1:500). Following three rinses with PBS, the slices were incubated with DAPI in PBS 0.1 M (Invitrogen, 5 mg/ml, 1:5000) for 5 min and rinsed again three to four times with PBS 0.1 M before being mounted on coverslips using an anti-fade mounting medium (Invitrogen Fluoro gold). After 24 h at 4 °C, slices were imaged on a confocal microscope (Leica LSM 710) and processed using ImageJ software.

**Chemicals**. Pilocarpine hydrochloride and γ-aminobutyric acid (GABA) was purchased from Sigma-Aldrich (Dorset, UK) and dissolved in saline on the day of experiment. 2,3-Dioxo-6-nitro-1,2,3,4-tetrahydrobenzo[f]quinoxaline-7-sulfonamide disodium salt (NBQX, 10 µM), DL-2-Amino-5-phosphonopentanoic (DL-AP5; 50 µM), picrotoxin (100 µM) and (2 S)-3-[[(1 S)-1-(3,4-Dichlorophenyl)ethyl]amino-2-hydroxypropyl](phenylmethyl) phosphinic acid (CGP55845; 1 µM), used in electrophysiological recordings in vitro to block AMPA, NMDA-, GABA_A- and GABA_B-receptors respectively, were all from Tocris Bioscience (Bristol, UK).

**Statistics**. All statistical data analyses were performed using IBM SPSS and OriginPro software, or algorithms coded in Python. Results are presented as mean ± s.e.m. Repeated one-way or two-way ANOVA, two-tailed unpaired and paired $t$-tests were used as appropriate. N values used in statistical analysis represent the number of animals in each experimental group. Statistical differences were considered significant at $P < 0.05$, and this was corrected for multiple comparisons using the post hoc Bonferroni correction.

**Reporting summary**. Further information on experimental design is available in the Nature Research Reporting Summary linked to this article.

**Data availability**
The data that support the findings of this study and the custom code for data analysis are available upon reasonable request. The source data underlying Figs. 2–5 and Supplementary Figures 1, 3–7 are provided as a Source Data file.

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

## Acknowledgements

This work was supported by Epilepsy Research UK (I.P. and D.M.K.), the Worshipful Company of Pewterers (Fellowship to I.P.), a Marie Sklodowska-Curie Actions Fellowship from the European Commission (A.L.), the Medical Research Council and the Wellcome Trust (D.M.K.). We are grateful to Z.J. Huang for making the *SOM::Cre* mice available.

## Author contributions

I.P. and D.M.K. conceived the study and designed experiments with V.M., V.M. performed the experiments and analysed the data, J.C. developed the state classifier, A.L. designed the KCC2 viral vector and optimised transgene overexpression protocols, D.M.K. and I.P. developed the FPGA-based closed-loop stimulation system, I.P. and D.M.K. wrote the manuscript, which was revised by all co-authors.

## Additional information

**Competing interests:** The authors declare no competing interests.

**Journal Peer Review Information:** *Nature Communications* thanks the anonymous reviewers for their contribution to the peer review of this work. Peer reviewer reports are available.

