## [Peer Review File · Nature Communications]

Reviewers' comments:

Reviewer #1 (Remarks to the Author):

The manuscript by Magloire et al. examines the effects of optogenetically manipulating PV interneurons on seizures elicited by pilocarpine application to cortex. The basic finding that the function of PV interneurons changes due to chloride accumulation is interesting. However, owing to the nature of the seizures, which are short and occur on an ongoing basis, the data are a bit complex to interpret, and some clarifying analyses would be helpful.

1. Several details of the experiment are a little unclear. First, my understanding is that seizures are detected, and then optogenetic stimulation is delivered if at least 20 sec has passed since the previous optogenetic stimulus was delivered. This ensures that the window preceding the stimulus has been free from optogenetic stimulation for at least 10 sec. Second, if there is a delay between seizure detection and optogenetic stimulation, then optogenetic stimulation is only delivered if the duration of the seizures exceeds that delay. Third, the duration of each seizure that triggers stimulation is compared to the immediately preceding one, unless the duration of the preceding seizure is less than the delay. In the latter case, the preceding seizure of requisite duration is used. Is all of this correct? If so, it might be helpful to spell it out more clearly in the results and in a diagram in the Figures.

2. It's not clear how the delay from seizure detection to optogenetic stimulation was implemented, i.e., were the delays varied randomly, interspersed, or did they progress sequentially? Were the delays varied during each subsequent seizure or were they varied in a block fashion. It would be ideal if these were interspersed, i.e., one delay did not consistently follow / precede another.

3. It's a bit odd to compare the effects of instantaneous vs. delayed optogenetic stimulation because in the former cases many seizures last < 2.5 sec whereas in the latter all are > 2.5 sec. So it's not really an apples to apples comparison. It would be better to restrict analysis to seizures which all last > 2.5sec, but this is also problematic because the effects of the optogenetic manipulation may be to reduce the duration of the seizures, potentially pushing more seizures below this threshold. This is an important issue though, and I suppose one way the authors could deal with it would be to plot the distribution of seizure durations for both conditions and include seizures which would have triggered stimulation after a delay, but did not do so because they were too short. This would make it easier to directly compare seizure durations in the two cases (instantaneous PV stimulation vs. stimulation after a delay >2.5 sec)

4. My understanding of the results is confused slightly because the duration of the seizures is very short and seizures are often occurring close together. Thus, there may be multiple seizures during the 10 second period of optogenetic stimulation. This raises some questions – what if the seizure duration goes down, but the frequency and/or duration of subsequent seizures during the period of illumination goes up? Such phenomena might call into question the interpretation that a reduction or prolongation of seizure duration corresponds to suppression or promotion of seizure activity. For these reasons, I think the author should at least analyze what happened to subsequent seizures during the period of light delivery. I realize it may be complicated to interpret these data, but I think it's important for the reviewers to see this to determine how it might fit in.

5. The experiments done to verify the effects of KCC2 overexpression are a bit odd because they were not done in perforated patch. Of course, the whole cell solution may not completely control the Cl concentration, but one should expect it to do so around the soma, which is where the majority of PV synapses should be located. These results would be more convincing if at least a few perforated patch recordings were also done.

6. What was the delay >2 sec – the exact value should be presented.

7. Why are raw durations shown in Figure 1, but then normalized durations only present in Figures 2 and 5? The raw durations should be presented in all cases.

Reviewer #2 (Remarks to the Author):

The first major claim of the manuscript is that the effects of optically-activated interneurons can change during a seizure event, in an awake behaving animal. The authors report that the particular timing of optogenetic activation of a major interneuron subpopulation (the PV-expressing interneurons) results in different effects upon seizure activity: early activation tends to reduce the duration of seizure activity, whilst later activation can increase the duration of seizure activity. This has previously been described in studies using brain slice preparations, but not in the awake brain. The authors then test a hypothesis around transient changes in intracellular chloride that has been well-supported by previous *in vitro* and *in vivo* studies. They test this hypothesis by modulating the principal neurons' chloride extrusion mechanisms by overexpressing the chloride co-transporter KCC2. This leads to the second major claim of the manuscript, which is that overexpression of KCC2 prevents the PV-expressing interneurons from having a seizure-extending effect, once the seizure has begun.

As the authors acknowledge, previous studies have shown that optically activating PV interneurons in awake mice can be anti-epileptic, and so this aspect of the manuscript reproduces previous work (e.g. Krook-Magnuson et al. 2013, Nat Comm). Meanwhile, previous *in vivo* studies have shown that intracellular chloride levels can change during intense periods of seizure activity (e.g. Sulis Sato et al. 2017, PNAS) and others have shown that KCC2 regulates intracellular chloride *in vivo* (e.g. Sivakumaran et al. 2015, J Neurosci; Chen et al. 2017, Sci Reports). The current paper extends previous work by bringing some of these ideas together in an awake brain preparation and offers some insight into why optically activating interneurons may produce complex effects. The novelty of the work is therefore based on the fact that the authors have conducted the majority of the work *in vivo*, using technically challenging awake behaving recordings and closed-loop photostimulation. The results have the potential to be interesting to scientists working on the cellular basis of epilepsy and those using optogenetic methods to disrupt seizure activity. The statistical analyses used in the manuscript are appropriate and a sufficient level of detail is generally provided for researchers to reproduce the work.

The work's strength is its use of *in vivo* recordings. However, there are several aspects of the work that would need to be addressed in order to make it convincing and significantly influence the thinking in the field:

(i) The study does not provide any direct evidence that the transmembrane chloride gradient is transiently collapsing during the seizure events. This would be challenging *in vivo*, but *in vitro* evidence under conditions that simulate the kinetics of the *in vivo* events would be an important demonstration. Providing data on the dynamics of the chloride may also help to explain counterintuitive aspects of the seizures, such as why short seizure intervals do not affect PV-mediated suppression, even though the neurons are thought to have loaded with chloride during the previous seizure (as shown in Fig. S3).

(ii) It is difficult to assess the data on the effects of *in vitro* and *in vivo* KCC2 overexpression as there is no control for lentiviral transduction.

(iii) The data does not provide definitive evidence that the overexpression of KCC2 changes the neurons' capacity to handle intracellular chloride. The manuscript includes two related types of data on this point. Using whole cell electrophysiology in acute slices, the authors first report that there is a less pronounced rundown of synaptically-evoked IPSCs in transduced cells. Second, they claim transduced neurons show less of a shift in intracellular chloride when they are subjected to a chloride loading protocol. Rather than reflecting differences in the cells' abilities to extrude chloride however, these effects could be explained by differences in how the cells are loading with chloride (presumably via the GABA-A receptors). This concern is evident in the first experiment, where the non-transduced example neuron has larger GABA-A currents than the transduced neuron, and so it would be predicted that the non-transduced neuron would load more rapidly. Meanwhile in the second experiment, the population data differ in how much their intracellular chloride levels change during the 'loading protocol' (i.e. the difference between transduced and non-transduced is evident immediately after the protocol). But the subsequent kinetics of chloride recovery appear similar in the two conditions. This would not be predicted if the rate of chloride clearance by KCC2 was different (e.g. Pellegrino et al. 2011, J Physiol). The authors should therefore exclude the possibility that these effects are related to differences in loading, which may be related to the expression of GABA-A receptors. Typically, this would be done by collapsing the intracellular chloride to the same level and then monitoring the rate of recovery.

(iv) To provide direct evidence for the proposed mechanism, it would be necessary to show experimentally that the KCC2 manipulation changes intracellular chloride dynamics during the sorts of seizure-like events that are observed here. Again, in vitro evidence under conditions that simulate kinetics of the in vivo events would be an important demonstration.

(v) The authors should address why the KCC2 overexpression does not increase the ability of PV interneurons to suppress the seizure events at short delays (e.g. 0 s data in Fig. 5F). This is important for the discussion on the therapeutic potential of augmenting KCC2 function.

Reviewer #3 (Remarks to the Author):

The authors present compelling evidence for the role of PV+ interneurons in sustaining seizures induced by the chemoconvulsant pilocarpine. Depending on the timing of the activation of the PV+, but not SOM+, interneurons the seizures could be curtailed or prolonged. The findings support the newly formed ideas about the role of perisomatic inhibition during seizure generation and maintenance. My major concern is about the presentation and analysis of the data. All data on the changes in seizure durations have been normalized, and it is not clear whether the paired analyses were performed on these normalized values. I suggest the following analysis on the raw data. Since ChR stimulation was delivered only 50% of the time, there must be as many seizures preceding seizures without stimulation as there are those with stimulation. The durations of the successive seizures can thus be plotted on an X-Y plot for both unstimulated and stimulated seizures. The slopes of the two linear regression can then be compared, and if there is a difference between the two slopes, it should mean that the stimulation altered seizure duration. I realize that the selection of long preceding seizures for the upcoming >2 s stimulations introduces a bias toward longer duration events, but this can also be applied to the events preceding the non-stimulated seizures. Analyzing the data in this manner should give an insight into the distribution of seizure durations, the random (or non-random) nature of successive seizure durations, and the unbiased effects of ChR or Arch stimulation, without the thwarting effects of normalization. There are also some other points that should be addressed:

- (1) How was the laser power used in the slice preparation equated with the in vivo power at the end of the optical fibre?
- (2) Suppl.Fig3 should indicate both horizontal and vertical error bars, or better yet, show all the individual data points from each of the 8 mice.
- (3) It is unclear whether the delays of ChR stimulation were randomized in a given mouse or different delays were used in different mice.
- (4) On l.151 the p-value appears to be significant, although the opposite is stated in the text,
- (5) The single trace effects shown in Suppl.Fig4 there should be supported by some quantification.
- (6) There are no viral controls for the KCC2 overexpression. It would have been preferable to use the transport-inactive KCC2 mutant Y1087D as control.
- (7) The discussion mentions the uneven subcellular distribution of KCC2, including its potential to influence the behavior of GABA-A receptors found at the AIS. It would be of interest to show whether the overexpressed KCC2 protein is also present at the AIS.

Below are our point-to-point responses to the reviewers' comments.

Reviewer 1:

1. *Several details of the experiment are a little unclear. First, my understanding is that seizures are detected, and then optogenetic stimulation is delivered if at least 20 sec has passed since the previous optogenetic stimulus was delivered. This ensures that the window preceding the stimulus has been free from optogenetic stimulation for at least 10 sec. Second, if there is a delay between seizure detection and optogenetic stimulation, then optogenetic stimulation is only delivered if the duration of the seizures exceeds that delay. Third, the duration of each seizure that triggers stimulation is compared to the immediately preceding one, unless the duration of the preceding seizure is less than the delay. In the latter case, the preceding seizure of requisite duration is used. Is all of this correct? If so, it might be helpful to spell it out more clearly in the results and in a diagram in the Figures.*

Yes, the reviewer is correct. We have clarified all these points by adding another schematic of the stimulation protocol to Supplementary Fig. 2.

2. *It's not clear how the delay from seizure detection to optogenetic stimulation was implemented, i.e., were the delays varied randomly, interspersed, or did they progress sequentially? Were the delays varied during each subsequent seizure or were they varied in a block fashion. It would be ideal if these were interspersed, i.e., one delay did not consistently follow / precede another.*

The delay was randomized in every experimental session in each animal, i.e. delays were arbitrarily varied avoiding consistent repetition and sequential progression. We have now clarified this in the text (p. 7).

3. *It's a bit odd to compare the effects of instantaneous vs. delayed optogenetic stimulation because in the former cases many seizures last < 2.5 sec whereas in the latter all are > 2.5 sec. So it's not really an apples to apples comparison. It would be better to restrict analysis to seizures which all last > 2.5sec, but this is also problematic because the effects of the optogenetic manipulation may be to reduce the duration of the seizures, potentially pushing more seizures below this threshold. This is an important issue though, and I suppose one way the authors could deal with it would be to plot the distribution of seizure durations for both conditions and include seizures which would have triggered stimulation after a delay, but did not do so because they were too short. This would make it easier to directly compare seizure durations in the two cases (instantaneous PV stimulation vs. stimulation after a delay >2.5 sec)*

Indeed, the direct comparison of seizure durations between the 0 s and >2 s delay groups is problematic. However, to address the reviewer's concern, we have further analysed our data and presented the paired comparisons for each seizure in each experimental delay group. In addition, we have also re-analysed our data excluding all seizures that were shorter than 2.5 s in the 0 s delay group. This exclusion does not change the anti-seizure effect of the immediate PV+ interneuron photostimulation. The results of these additional analyses are now presented on p. 8 and in Supplementary Fig. 4c and d.

4. *My understanding of the results is confused slightly because the duration of the seizures is very short and seizures are often occurring close together. Thus, there may be multiple seizures during the 10 second period of optogenetic stimulation. This raises some questions – what if the seizure duration goes down, but the frequency and/or duration of subsequent seizures during the*

period of illumination goes up? Such phenomena might call into question the interpretation that a reduction or prolongation of seizure duration corresponds to suppression or promotion of seizure activity. For these reasons, I think the author should at least analyze what happened to subsequent seizures during the period of light delivery. I realize it may be complicated to interpret these data, but I think it's important for the reviewers to see this to determine how it might fit in.

We have now analysed the number of ictal discharges in 10 s intervals before, during and after photostimulation and added this to Fig. 2 (panel f). PV+ activation had no significant effect on the number of events.

5. *The experiments done to verify the effects of KCC2 overexpression are a bit odd because they were not done in perforated patch. Of course, the whole cell solution may not completely control the Cl concentration, but one should expect it to do so around the soma, which is where the majority of PV synapses should be located. These results would be more convincing if at least a few perforated patch recordings were also done.*

Here we are interested in comparing the chloride extrusion capacity between the control and KCC2 over-expression groups. We have previously demonstrated that the whole-cell configuration is sufficiently sensitive to detect differences in both chloride extrusion rates and E_{Cl} produced by the KCC2-specific mutations (Stöberg et al., 2015, Mutations in SLC12A5 in epilepsy of infancy with migrating focal seizures. 6:8038, *Nat. Comm.*) even in HEK cells, producing results similar to those obtained in the perforated patch experiments (Saito et al., 2016, Impaired neuronal KCC2 function by biallelic SLC12A5 mutations in migrating focal seizures and severe developmental delay. 6:30072, *Scientific Reports*). We therefore feel that replicating our findings in the perforated patch configuration would not provide any additional information, particularly, taking into account the results of additional experiments with the halorhodopsin chloride loading (Fig. 4, see below).

6. *What was the delay >2 sec – the exact value should be presented.*

This is 3.63 ± 0.26 s for the PV-ChR2 group. We have now added the exact values for the >2s delays in the PV-ChR2 and PV-ArchT experiments on p. 7 and 9, respectively. The rest of the values are presented in the Materials and Methods along with the p values for their comparison, showing that there are no statistically significant differences between the groups (p. 20).

7. *Why are raw durations shown in Figure 1, but then normalized durations only present in Figures 2 and 5? The raw durations should be presented in all cases.*

We present the normalised data in Fig. 3 and 5 to demonstrate the evolution of the effect of PV+ activation that occurs as the delay of photostimulation increases. We have now also provided all raw durations related to Figs. 3 and 5 in scatter plots in Supplementary Figs. 4 and 8 (as suggested by Reviewer 3, see below). We have also provided all related statistical comparisons.

Reviewer 2:

(i) *The study does not provide any direct evidence that the transmembrane chloride gradient is transiently collapsing during the seizure events. This would be challenging in vivo, but in vitro evidence under conditions that simulate the kinetics of the in vivo events would be an important demonstration. Providing data on the dynamics of the chloride may also help to explain counterintuitive aspects of the seizures, such as why short seizure intervals do not affect PV-mediated suppression, even though the neurons are thought to have loaded with chloride during the previous seizure (as shown in Fig. S3).*

In vitro measurement of intracellular chloride during seizures has already been reported by Raimondo et al. (*Frontiers in Cellular Neuroscience*, 7:202, 2013). Using a genetically-encoded ratiometric chloride and pH sensor, ClopHensorN, they showed that the chloride concentration in pyramidal neurons rapidly increases as early as 2 s after the seizure onset in the hippocampal slices. Characterisation of intracellular chloride dynamics during seizures *in vivo* would of course be invaluable, but we do not think that repeating these slice experiments would add value to our study of the role of interneurons in controlling seizures in awake animals. We have now updated the discussion in the light of these points on p. 13.

(ii) *It is difficult to assess the data on the effects of in vitro and in vivo KCC2 overexpression as there is no control for lentiviral transduction.*

We agree, and have now performed additional experiments using a control CaMKII-GFP lentivirus for the *in vivo* experiment, and a control CaMKII-Halo-GFP for the chloride loading experiments *in vitro*. The results strengthen the conclusions of the study and are presented in Figures 4 and 5.

(iii) *The data does not provide definitive evidence that the overexpression of KCC2 changes the neurons' capacity to handle intracellular chloride. The manuscript includes two related types of data on this point. Using whole cell electrophysiology in acute slices, the authors first report that there is a less pronounced rundown of synaptically-evoked IPSCs in transduced cells. Second, they claim transduced neurons show less of a shift in intracellular chloride when they are subjected to a chloride loading protocol. Rather than reflecting differences in the cells' abilities to extrude chloride however, these effects could be explained by differences in how the cells are loading with chloride (presumably via the GABA-A receptors). This concern is evident in the first experiment, where the non-transduced example neuron has larger GABA-A currents than the transduced neuron, and so it would be predicted that the non-transduced neuron would load more rapidly. Meanwhile in the second experiment, the population data differ in how much their intracellular chloride levels change during the 'loading protocol' (i.e. the difference between transduced and non-transduced is evident immediately after the protocol). But the subsequent kinetics of chloride recovery appear similar in the two conditions. This would not be predicted if the rate of chloride clearance by KCC2 was different (e.g. Pellegrino et al. 2011, J Physiol). The authors should therefore exclude the possibility that these effects are related to differences in loading, which may be related to the expression of GABA-A receptors. Typically, this would be done by collapsing the intracellular chloride to the same level and then monitoring the rate of recovery.*

We did not systematically explore the difference in the magnitude of the IPSCs in the transduced and non-transduced neurons. However, as the reviewer correctly points out, the difference between the KCC2 overexpressing and control, non-transduced neurons may become evident during the loading process. This could explain why the transduced neurons displayed a less prominent increase in the IPSC amplitude immediately after the loading protocol in the first place.

We recognise the reviewer's concern that the unequal chloride loading may potentially be explained by differences in GABA_A receptor expression. To exclude this possibility, we took an alternative approach bypassing GABA_A activation during the loading. In this new set of experiments, we loaded neurons with chloride by activating the light-sensitive chloride pump Halorhodopsin, as described previously (Raimondo et al. 2012). Again, transduced neurons displayed faster recovery of the GABA_A receptor-mediated IPSCs to the baseline level. We have included these new data instead of experiments with GABA puff-induced chloride loading in Fig. 4.

(iv) *To provide direct evidence for the proposed mechanism, it would be necessary to show experimentally that the KCC2 manipulation changes intracellular chloride dynamics during the sorts of seizure-like events that are observed here. Again, in vitro evidence under conditions that simulate kinetics of the in vivo events would be an important demonstration.*

We agree that further characterisation of the link between KCC2 activity and the dynamics of intracellular chloride during seizures would be useful. The only realistic way to address this would be to perform chloride imaging experiments during seizures while manipulating KCC2 activity. However, given the limitations of the current chloride imaging tools (e.g. the pH sensitivity of the chloride sensors) such *in vivo* experiments would be almost impossible to interpret, and further *in vitro* experiments would have limited additional value for this study.

(v) *The authors should address why the KCC2 overexpression does not increase the ability of PV interneurons to suppress the seizure events at short delays (e.g. 0 s data in Fig. 5F). This is important for the discussion on the therapeutic potential of augmenting KCC2 function.*

We agree that this is an important point for the discussion and have updated the text on p. 12 in the results section as well as on p. 14 in the discussion.

Reviewer 3:

My major concern is about the presentation and analysis of the data. All data on the changes in seizure durations have been normalized, and it is not clear whether the paired analyses were performed on these normalized values. I suggest the following analysis on the raw data. Since ChR stimulation was delivered only 50% of the time, there must be as many seizures preceding seizures without stimulation as there are those with stimulation. The durations of the successive seizures can thus be plotted on an X-Y plot for both unstimulated and stimulated seizures. The slopes of the two linear regression can then be compared, and if there is a difference between the two slopes, it should mean that the stimulation altered seizure duration. I realize that the selection of long preceding seizures for the upcoming >2 s stimulations introduces a bias toward longer duration events, but this can also be applied to the events preceding the non-stimulated seizures. Analyzing the data in this manner should give an insight into the distribution of seizure durations, the random (or non-random) nature of successive seizure durations, and the unbiased effects of ChR or Arch stimulation, without the thwarting effects of normalization.

We have now added the raw data and statistics related to Figures 3 and 5 in Supplementary Fig. 4 and 8. As suggested by the reviewer, we have presented the individual paired seizures (control vs stimulated) on a scatter plot for each delay (Supplementary Fig. 4 and 8).

(1) *How was the laser power used in the slice preparation equated with the in vivo power at the end of the optical fibre?*

We matched the power at the tip of the optic fibre and objective of the microscope. This point is now clarified on p. 6.

(2) *Suppl.Fig3 should indicate both horizontal and vertical error bars, or better yet, show all the individual data points from each of the 8 mice.*

We have now added both horizontal and vertical error bars in Supplementary Fig 3.

(3) *It is unclear whether the delays of ChR stimulation were randomized in a given mouse or different delays were used in different mice.*

Stimulation was randomized in each mouse. We have now added a sentence to clarify this on p. 7.

(4) *On l.151 the p-value appears to be significant, although the opposite is stated in the text,*

We apologise for the confusing sentence; we have re-worded this.

(5) *The single trace effects shown in Suppl.Fig4 there should be supported by some quantification.*

We have added the required information to the Supplementary Fig. 5.

(6) *There are no viral controls for the KCC2 overexpression. It would have been preferable to use the transport-inactive KCC2 mutant Y1087D as control.*

We have performed additional control experiments using lentiviral CaMKII-GFP (Fig. 4 and 5; see also response to Reviewer 2).

(7) *The discussion mentions the uneven subcellular distribution of KCC2, including its potential to influence the behavior of GABA-A receptors found at the AIS. It would be of interest to show whether the overexpressed KCC2 protein is also present at the AIS.*

Although we agree with the reviewer that this would be of interest, we feel that assessing the distribution of the over-expressed KCC2 protein is beyond the scope of this paper. It would be very interesting to over-express KCC2 in the dendritic, somatic and axonal compartments and see how this affects the action of different interneurons in the future experiments. We have now added this point to the discussion (p. 15).

Reviewers' comments:

Reviewer #1 (Remarks to the Author):

Overall the reviewers have done an excellent job of responding to the issues I raised. I have one more minor recommendation. Supplementary Fig S4 is really great it really shows the main finding in a very convincing way. I would strongly recommend making this a main figure.

Reviewer #2 (Remarks to the Author):

In reference to the points in the original review:

(i) The revised manuscript still does not provide any direct evidence on how the key parameter in the authors' explanatory hypothesis - intracellular chloride - is changing during the seizure events that are being studied. This is important because compared to previous work, the seizure events here are of short duration (typically 5s; Table 1), and are occurring at high frequency (typical seizure intervals of 2-12 s; Fig S3). Previous work examining seizure-associated chloride dynamics has examined more sustained seizure events that recruit large network activity and are more spaced in time. If I understand correctly, the proposed model is that during these short seizures the neurons load sufficient chloride so that PV inputs switch from being initially inhibitory, to becoming excitatory, and then the neurons recover their chloride before the next seizure, so that PV inputs have an inhibitory action at the start of the next seizure event. The new data in Figure 4 shows that chloride extrusion mechanisms operate on a slow timescale (time constant of approx. 47 seconds) and it looks like the loading of the chloride was achieved over a 1 minute period. These are very different to what is being proposed for the seizures. For example, if there is chloride loading during each seizure, and extrusion operates with this sort of time constant, it is difficult to imagine how the chloride is not accumulating across seizures. If the authors are not able to make direct measurements of chloride, they should discuss their interpretation of the data in the context of the temporal properties of their seizure activity and how this relates to the kinetics of intracellular chloride regulation.

(ii) In the revised version of the manuscript, the authors have made the important addition of an appropriate control virus.

(iii) The revised manuscript includes a well-controlled experiment to show that KCC2 transduced cells have accelerated rates of chloride extrusion (faster by factor of 3 from a time constant of approx. 47s to 16s). This data is compelling in showing that the KCC2 overexpression can accelerate chloride recovery.

(iv) Corrected measurements of seizure-related chloride dynamics have been conducted in vivo (Sulis Sato et al. 2017, PNAS). The authors are correct that this is challenging. Please see related points above.

(v) The authors have added two sentences in the Discussion. It would seem important to also discuss the very recent paper from the Moss group, which reaches the opposite conclusion about how increased KCC2 activity can affect seizures (Moore et al. 2018, PNAS).

Reviewer #3 (Remarks to the Author):

I am satisfied with the changes.

Below are our point-to-point responses to the reviewers' comments.

Reviewer #1 (Remarks to the Author):

Overall the reviewers have done an excellent job of responding to the issues I raised. I have one more minor recommendation. Supplementary Fig S4 is really great it really shows the main finding in a very convincing way. I would strongly recommend making this a main figure.

We thank the reviewer for this suggestion and have now incorporated part of the Supplementary Fig S4 into the main Fig 3.

Reviewer #2 (Remarks to the Author):

1 The revised manuscript still does not provide any direct evidence on how the key parameter in the authors' explanatory hypothesis - intracellular chloride - is changing during the seizure events that are being studied. This is important because compared to previous work, the seizure events here are of short duration (typically 5s; Table 1), and are occurring at high frequency (typical seizure intervals of 2-12 s; Fig S3). Previous work examining seizure-associated chloride dynamics has examined more sustained seizure events that recruit large network activity and are more spaced in time. If I understand correctly, the proposed model is that during these short seizures the neurons load sufficient chloride so that PV inputs switch from being initially inhibitory, to becoming excitatory, and then the neurons recover their chloride before the next seizure, so that PV inputs have an inhibitory action at the start of the next seizure event. The new data in Figure 4 shows that chloride extrusion mechanisms operate on a slow timescale (time constant of approx. 47 seconds) and it looks like the loading of the chloride was achieved over a 1 minute period. These are very different to what is being proposed for the seizures. For example, if there is chloride loading during each seizure, and extrusion operates with this sort of time constant, it is difficult to imagine how the chloride is not accumulating across seizures. If the authors are not able to make direct measurements of chloride, they should discuss their interpretation of the data in the context of the temporal properties of their seizure activity and how this relates to the kinetics of intracellular chloride regulation.

Our *in vitro* experiments were designed to test whether we would be able to increase the ability of principal neurons to extrude chloride by transducing them with our KCC2 construct. Indeed, we found that KCC2 overexpressing neurons had ~3-fold faster chloride extrusion rates compared to non-transduced ones. However, we would be reluctant to use the time constants obtained in these experiments as estimates of the temporal profile of the chloride gradients during *in vivo* seizures. There are two important points worth considering in this respect: the difference between whole-cell recordings *in vitro* and extracellular recordings *in vivo*, and the relative amount of chloride loading in two situations.

First, KCC2 activity is tightly regulated by various intracellular signalling pathways, including those that are sensitive to intracellular chloride itself (e.g. SPAK and OSR1 kinases) (Kahle et al., *Trends Mol Med*, 2015). Therefore, disruption of some of these mechanisms in whole-cell recordings means that measuring the relaxation of GABA_AR-mediated currents may not accurately reflect the rate of chloride extrusion during seizures *in vivo*. Second, although the temporal profile of intracellular ionic shifts *in vivo* is yet to be established, *in vitro* a seizure lasting 30-60 s can increase intracellular chloride by ~20 mM (Raimondo et al, *Front Cell Neurosci*, 2013). Furthermore, Sulis Sato et al (*PNAS*, 2017) showed ~8mM chloride being extruded over a 10 s interval following seizure termination *in vivo*. This could easily translate into several tens of seconds following a minute-long chloride load *in vitro* - explaining the slow time constants in the experiments shown in Figure 4. In our *in vivo* experiments, however, the seizures were much shorter (even longest not exceeding 10s),

and the average seizure frequency was 0.18 Hz (Fig 2e,f). Whilst, in principle, slow recovery of the intracellular chloride concentration after a seizure might impact on the next ictal event, in our experiments, photostimulation of PV+ cells at the onset of seizures had a reliable anti-seizure effect, suggesting that the inter-ictal intervals were sufficient to restore the inhibitory power of these interneurons.

As suggested by the reviewer, we have now addressed these points in the discussion on pages 13-14.

2 In the revised version of the manuscript, the authors have made the important addition of an appropriate control virus.

3 The revised manuscript includes a well-controlled experiment to show that KCC2 transduced cells have accelerated rates of chloride extrusion (faster by factor of 3 from a time constant of approx. 47s to 16s). This data is compelling in showing that the KCC2 overexpression can accelerate chloride recovery.

4 Corrected measurements of seizure-related chloride dynamics have been conducted in vivo (Sulis Sato et al. 2017, PNAS). The authors are correct that this is challenging. Please see related points above.

We thank the reviewer for pointing out this study. We noted that the most prominent changes in the intracellular chloride (~6mM over 4s) were associated with the temporal evolution of fast oscillations of LFP and rapidly decreased after those stopped. We have now mentioned this paper in our revised discussion on pages 13-14.

5 The authors have added two sentences in the Discussion. It would seem important to also discuss the very recent paper from the Moss group, which reaches the opposite conclusion about how increased KCC2 activity can affect seizures (Moore et al. 2018, PNAS).

We already cited this paper in the manuscript (reference 37). The conclusions of this study are somewhat different from ours, but not the opposite. Both studies demonstrate that potentiation of KCC2 function may reduce seizures. Moore et al. showed that increased KCC2 activity reduced the frequency and duration of epileptiform bursts *in vitro* and limited the severity of seizures in an *in vivo* kainate model. While we did not explore the impact of KCC2 overexpression on seizure susceptibility, our experiments demonstrate that interneuron-mediated seizure prolongation is abolished by overexpressing KCC2.

We would also like to note that altered regulation of co-transporter activity caused by a mutation is functionally different from overexpressing the protein; thus the outcomes of these two manipulations may differ.

We have now discussed this paper in more detail on page 15.

Reviewer #3 (Remarks to the Author):

I am satisfied with the changes.

We thank the reviewer for their constructive feedback and comments.

REVIEWERS' COMMENTS:

Reviewer #1 (Remarks to the Author):

Overall the reviewers have done an excellent job of responding to the issues I raised. I have one more minor recommendation. Supplementary Fig S4 is really great it really shows the main finding in a very convincing way. I would strongly recommend making this a main figure.

Reviewer #2 (Remarks to the Author):

The authors have extended their discussion to highlight some unanswered questions regarding the kinetics of chloride handling and the challenges of moving between in vitro and in vivo models. The authors also relate their findings to the paper from the Moss group. This makes for an improved discussion that reflects the current literature.

Reviewer #3 (Remarks to the Author):

I am satisfied with the changes.

Reviewer #1 (Remarks to the Author):

Overall the reviewers have done an excellent job of responding to the issues I raised. I have one more minor recommendation. Supplementary Fig S4 is really great it really shows the main finding in a very convincing way. I would strongly recommend making this a main figure.

We have merged Fig. S4 and Fig. 3 as suggested.

Reviewer #2 (Remarks to the Author):

The authors have extended their discussion to highlight some unanswered questions regarding the kinetics of chloride handling and the challenges of moving between in vitro and in vivo models. The authors also relate their findings to the paper from the Moss group. This makes for an improved discussion that reflects the current literature.

No changes requested

Reviewer #3 (Remarks to the Author):

I am satisfied with the changes.

No changes requested